

# Inhibitory effect of carvedilol on bedaquiline metabolism *in vitro* and *in vivo*

Qingqing Li[1,*], Wanshu Li[1,*], Jie Chen[2], Hangjuan Lin[1] and Cixia Zhou[1]

[1] Ningbo Municipal Hospital of Traditional Chinese Medicine (TCM), Affiliated Hospital of Zhejiang Chinese Medical University, Ningbo, China
[2] The First Affiliated Hospital of Wenzhou Medical University, Wenzhou, China
[*] These authors contributed equally to this work.

## ABSTRACT

Bedaquiline has recently been approved for the treatment of multidrug-resistant tuberculosis. Carvedilol is a cardiovascular medication extensively used in the treatment of heart failure and hypertension. In this study, Sprague-Dawley rats, rat liver microsomes (RLM), human liver microsomes (HLM), and recombinant human CYP3A4 were used to explore the effect of carvedilol on the metabolism of bedaquiline. Ultra-performance liquid chromatography-tandem mass spectrometry was used to facilitate the quantification of the analyte concentrations. *In vitro,* carvedilol did not exhibit time-dependent inhibition of bedaquiline, which aligns with the half-maximal inhibitory concentration ($IC_{50}$) shift results. The $IC_{50}$ values of carvedilol were $15.35 \pm 0.43\,\mu M$ in RLM, $7.55 \pm 0.74\,\mu M$ in HLM, and $0.79 \pm 0.05\,\mu M$ in CYP3A4. Besides, the inhibition type of carvedilol was found to be mixed, un-competitive, and mixed in RLM, HLM, and CYP3A4, respectively. *In vivo*, the co-administration of carvedilol with bedaquiline resulted in a significant increase in the area under the plasma concentration-time curve $AUC_{(0-t)}$, $AUC_{(0-\infty)}$, and $C_{max}$ of bedaquiline while decreasing its $CL_{z/F}$.
**Lay summary**: Carvedilol could inhibit the metabolism of bedaquiline *in vitro* and *in vivo*, with different mechanisms in different enzymatic reaction systems. Hence, caution should be exercised when combining bedaquiline with carvedilol.

## INTRODUCTION

Multidrug-resistant tuberculosis (MDR-TB) is a growing global concern, as rising levels of resistance lead to soaring death rates. Bedaquiline was approved by the US Food and Drug Administration (FDA) in 2012 for treating pulmonary MDR-TB, which became one of the important drugs in the treatment of long-term MDR-TB (*Kotwal et al., 2020*). Nevertheless, studies have demonstrated that bedaquiline may cause adverse effects such as cardiac toxicity (*Worley & Estrada, 2014*), hepatic toxicity (*Kakkar & Dahiya, 2014*), and phospholipidosis (*Diacon et al., 2012*; *Guillemont et al., 2011*). Patients with TB are usually treated with concurrent multiple-drug therapy; therefore, the risk of drug–drug interactions (DDIs) also increases rapidly. In 1909, Pottenger proposed that TB affected

Corresponding authors
Hangjuan Lin, nbszyy_lhj@126.com
Cixia Zhou, zcx_nbszyy@126.com

the cardiovascular system (CVS) (*Pottenger, 1909*). A subsequent study by *Levinsky (1961)* further demonstrated that TB could predispose individuals to cardiopulmonary failure. In recent years, *Ohene et al. (2019)* have also revealed that patients with TB and CVS complications exhibited a mortality rate approaching 60%. Moreover, DDIs are the main reason for the stratification of drug efficacy. Therefore, in this study, we systematically screened a series of drugs with bedaquiline.

The drug–drug interactions may be due to the inhibition or induction of drug metabolism enzymes and drug transporter protein levels. Bedaquiline is mainly metabolized to a less active metabolite (M2) through CYP3A4 (*Liu et al., 2014*) and a substrate of *p*-glycoprotein (*p*-gp) (*Kotwal et al., 2020*). CYP3A4 is the most important member of the cytochrome P450 superfamily, metabolizing approximately 30%–40% of clinical drugs, such as analgesics (tramadol), antidepressants (quetiapine), and antimicrobials (erythromycin) (*Zanger et al., 2008*). When bedaquiline was combined with a CYP3A4 inhibitor (fluconazole), its $C_{max}$ and area under the plasma concentration-time curve $(AUC)_{(0-t)}$ increased by 45% and 46%, respectively, and when combined with a CYP3A4 inducer (carbamazepine), its $C_{max}$ and $AUC_{(0-t)}$ decreased by 13% and 16%, respectively (*Kotwal et al., 2020*). Accordingly, we should take necessary precautions to avoid the combination of bedaquiline with drugs that may inhibit or induce CYP3A4.

In the present study, ultra-performance liquid chromatography-tandem mass spectrometry (UPLC-MS/MS) was used to quantify the concentration of analytes. Fifty drugs were selected to investigate their influence on the metabolism of bedaquiline *in vitro*. Additionally, the mechanism of carvedilol, which exhibited the strongest inhibitory activity, was specifically explored in rat liver microsomes (RLM), human liver microsomes (HLM), and recombinant human CYP3A4. We also evaluated the effect of carvedilol on bedaquiline pharmacokinetics in rats *in vivo*.

## MATERIALS AND METHODS

### Chemicals and reagents

Bedaquiline, M2, and fluconazole (internal standard, IS) were bought from Yingxin Biotechnology Co., Ltd. (Shanghai, China). Various drugs (andrographolide, amiodarone hydrochloride, apigenin, apixaban, artemether, astragalin, baicalin, berberine, bergenin, betaine, bosentan, carvedilol, cangrelor, cariprazine, chrysin, cimetidine, dabigatran, daidzein, daphnoretin, genistein, glimepiride, hesperetin, isorhamnetin, kaempferol, limonin, imipramine hydrochloride, losartan, lovastatin, lycopene, mexiletine hydrochloride, naringenin, naringin, nifedipine, quercetin, quinidine, PF-04971729, piperine, propafenone, resveratrol, rivaroxaban, rutin, selexipag, silibinin, sitagliptin, sophocarpine, sophoridine, verapamil hydrochloride, vericiguat, warfarin, 2-(4-hydroxyphenyl) acetic acid) were provided by Shanghai Chuangsai Technology Co., Ltd. (Shanghai, China). RLM and HLM were prepared by our team (*Wang et al., 2015*). Recombinant human CYP3A4 was obtained from the Beijing Hospital (Beijing, China). Reduced nicotinamide adenine dinucleotide phosphate (NADPH) was obtained from Roche Pharmaceutical, Ltd. (Basel, Switzerland). The remaining chemicals and reagents were of analytical grade.

## Equipment and operation conditions

The concentrations of bedaquiline and its metabolite (M2) were detected using UPLC-MS/MS, which was equipped with a Waters Acquity UPLC system (Milford, MA, USA) and a Waters Xevo TQS triple quadrupole tandem mass spectrometer (Milford, MA, USA). Chromatographic separations of the analytes were performed on the Bridged Ethylene Hybrid (BEH) C18 column (2.1 × 50 mm, 1.7 μm; Milford, MA, USA), and it was operated at a column temperature of 40 °C. The mobile phases were acetonitrile (ACN, a) and 0.1% formic acid (b) at 0.30 mL/min gradient elution for 2.0 min. The gradient conditions were as follows: 90% B (0–0.5 min), 90%–10% B (0.5–1.0 min), 10% B (1.0–1.4 min), 10%–90% B (1.4–1.5 min), and the last 90% B (1.5–2.0 min) was maintained for balance. The temperature of the auto-sample injector was set at 4 °C, and the sample volume was 1.0 μL for each run. Mass spectral information of the analytes was obtained by selective response monitoring in the positive ion mode. To improve the sensitivity and specificity of the analysis, the transitions of bedaquiline, M2, and IS were $m/z$ 555.00 > 58.08, $m/z$ 540.93 > 479.92, and $m/z$ 307.14 > 238.14, respectively.

### In vitro experiments

200 μL enzymatic reaction system contained 1.0 M phosphate buffered saline (PBS), 1.0 mM NADPH, 0.3 mg/mL RLM or 0.3 mg/mL HLM or 0.5 pmol CYP3A4, 0–100 μM bedaquiline to determine the $K_m$ (Michaelis–Menten constant) values of bedaquiline. The mixture without NADPH was pre-incubated at 37 °C for 5 min, and then 20 mM NADPH was added to activate the reaction. After incubation for 30 min, the reaction was stopped by rapid cooling to −80 °C. Next, 500 μL ACN and 20 μL IS (200 ng/mL) were added to precipitate protein before the samples melted. The mixture was vortexed for 2 min and centrifuged at 13,000 rpm for 10 min. Finally, 100 μL supernatant was collected for UPLC-MS/MS analysis (Lin et al., 2019).

To detect the possible DDIs with bedaquiline, the 200 μL system was kept the same, and the $K_m$ value was used as the concentration of bedaquiline in the RLM system to determine the inhibitory effect of 50 kinds of drugs (the concentration of each drug as the inhibitor was 100 μM) on bedaquiline metabolism.

To explore the half-maximal inhibitory concentration ($IC_{50}$), the concentrations of carvedilol were set at 0, 0.01, 0.1, 1, 10, 25, 50, and 100 μM, while bedaquiline was set at 10 μM in RLM, 80 μM in HLM, and 0.8 μM in CYP3A4, according to their $K_m$ values, respectively. The incubation system consisted of 0.3 mg/mL RLM or 0.3 mg/mL HLM or 0.5 pmol CYP3A4, 1.0 mM NADPH, bedaquiline, carvedilol, and 1.0 M phosphate-buffered saline buffer to the final volume of 200 μL.

To investigate the potential inhibitory mechanism, the concentrations of carvedilol and bedaquiline were determined based on $IC_{50}$ and $K_m$ values, respectively. Carvedilol's concentrations were 0, 7.5, 15, and 30 μM in RLM, 0, 1.9, 3.8, and 7.6 μM in HLM, and 0, 0.4, 0.8, and 1.6 μM in CYP3A4, while the concentrations of bedaquiline were 2.5, 5, 10, and 20 μM or 20.8, 41.5, 83, and 166 μM or 2.5, 5, 10, and 20 μM in RLM or HLM or CYP3A4, respectively. The sample was prepared following the experiment mentioned above.

To study the time-dependent inhibition (TDI), the incubation system was the same as that described for detecting possible drug interactions with bedaquiline. The mixture with or without NADPH was incubated at 37 °C for 30 min, where the concentration of carvedilol was set as 0, 0.01, 0.1, 1, 10, 25, 50, and 100 µM. Thereafter, bedaquiline (the corresponding $K_m$ values: 10 µM in RLM, 80 µM in HLM, and 0.8 µM in CYP3A4) was added and incubated for another 30 min. The following procedures were the same as the experiment above.

### *In vivo* experiments

Sprague-Dawley (SD) male rats (200 ± 10 g) were provided by the Animal Experimental Center of The First Affiliated Hospital of Wenzhou Medical University (Zhejiang, China) and were housed in an animal experimental center with a temperature range of 20–26 °C, relative humidity of 55% ± 15%, and a 12 h/day light/dark cycle. Before the experiment, the rats were fasted for 12 h and allowed free access to water. All animal experiments followed the National Research Council's Guide for the Care and Use of Laboratory Animals. Furthermore, the study protocol followed the ARRIVE guidelines and was approved by the Laboratory Animal Ethics Committee of The First Affiliated Hospital of Wenzhou Medical University (Ethics approval number: WYYY-IACUC-AEC-2023-046).

Eight SD rats were randomly divided into two groups ($n = 4$): a single group (group A) and a combined group (group B). Bedaquiline and carvedilol were dissolved with 0.5% carboxymethylcellulose sodium salt (CMC-Na) solution to the desired concentrations. Group B was orally administered carvedilol (5 mg/kg), while group A administered an equivalent volume of 0.5% CMC-Na solution by gavage. Thirty minutes later, both groups administered oral doses of bedaquiline (10 mg/kg) (*Kotwal et al., 2020*). The time points of the collected tail vein blood samples were 0.5, 1, 2, 3, 4, 6, 8, 10, 12, 24, and 48 h after bedaquiline administration. Finally, conform to the AVMA guidelines for the euthanasia of animals, eight SD rats were given 2% isoflurane inhalation anesthesia followed by 5% isoflurane concentration. Plasma (100 µL) was mixed with 300 µL ACN and 20 µL IS (200 ng/mL), vortexed, and centrifuged thoroughly, and the supernatant (100 µL) was obtained for UPLC-MS/MS detection.

### Statistical analysis

The Michaelis–Menten, $IC_{50}$, Lineweaver–Burk plots and mean plasma concentration–time curves were plotted by GraphPad Prism software (version 9.5; GraphPad Software Inc., CA, USA). The pharmacokinetic parameters were obtained by non-compartmental analysis using Drug and Statistics software (version 3.0; Shanghai University of Traditional Chinese Medicine, Shanghai, China). All data were expressed as mean ± standard deviation (SD) and analyzed statistically using the Statistical Package for the Social Sciences software (version 24.0; SPSS Inc., Chicago, IL, United States). $P < 0.05$ was considered statistically significant.

## RESULTS

### UPLC-MS/MS analysis

The UPLC-MS/MS chromatograms of bedaquiline, M2, and IS (fluconazole) under three different conditions are presented in Fig. 1. The retention times of bedaquiline, M2, and IS were 1.36, 1.35, and 1.17 min, respectively. No endogenous interferences were observed to affect the determination of bedaquiline, M2, and IS. Moreover, the precision and accuracy of this method were both less than 15%. The concentration range of the calibration standard curve of bedaquiline was 1–5,000 ng/mL, and M2 was 1–500 ng/mL, with correlation coefficients over 0.999. The lower limit of quantification (LLOQ) of bedaquiline and M2 was 1 ng/mL. In addition, the stability, matrix effect (ME), and extraction recovery of samples in this method were all accorded with the requirements.

### Carvedilol inhibited the metabolism of bedaquiline *in vitro*

According to Fig. 2, bedaquiline was a self-inhibiting substrate, and carvedilol exhibited the most powerful restraining power of 50 drugs, with 81.88% inhibition of bedaquiline. Then, combined with the result of $IC_{50}$, carvedilol strongly inhibited the metabolism of bedaquiline, with $IC_{50}$ values of $15.35 \pm 0.43$ $\mu$M in RLM, $7.55 \pm 0.74$ $\mu$M in HLM, and $0.79 \pm 0.05$ $\mu$M in CYP3A4, respectively (Fig. 3). Therefore, we next conducted to explore the underlying mechanisms. The $IC_{50}$ shift was used to assess the existence of TDI. Commonly, if the decrease in enzyme activity in the presence of NADPH (+NADPH) was greater than that in the absence of NADPH (−NADPH), TDI was suspected, and an $IC_{50}$ shift fold >10 was a time-dependent inhibition (*Jin et al., 2015*). As exhibited in Fig. 4, $IC_{50(-NADPH)}/IC_{50(+NADPH)}$ was 1.18, 1.92, and 5.00 for RLM, HLM, and CYP3A4, respectively. Moreover, the Lineweaver–Burk plot intersected in the second quadrant in RLM, exhibited a set of parallel lines in HLM, and intersected in the third quadrant of CYP3A4 (Fig. 5). Accordingly, these data demonstrated that carvedilol inhibited the metabolism of bedaquiline was all not in a time-dependent inhibition, and the inhibition type was mixed of non-competitive + competitive, un-competitive, and mixed of non-competitive + un-competitive, with the $\alpha K_i$ values of 61.32, 8.44, and 0.13 in RLM, HLM, and CYP3A4, respectively (Table 1).

### Carvedilol changed the main pharmacokinetic parameters of bedaquiline *in vivo*

The mean concentration–time curves of bedaquiline and M2 were demonstrated in Fig. 6. Based on Tables 2 and 3, we found that carvedilol could raise the $AUC_{(0-t)}$, $AUC_{(0-\infty)}$, and $C_{max}$ values of bedaquiline by 2.15–2.28 fold and declined the $CL_{z/F}$ of bedaquiline by about 4.33 fold. Furthermore, carvedilol prolonged the $T_{max}$ of bedaquiline while shortening its $t_{1/2z}$. However, there was a non-significant difference in M2 when bedaquiline was used in combination with carvedilol compared to the single-used.

## DISCUSSION

TB is caused by Mycobacterium tuberculosis (*Deshkar & Shirure, 2022*), with an incidence rate of approximately 500,000 cases and an estimated 2–3 million deaths worldwide each

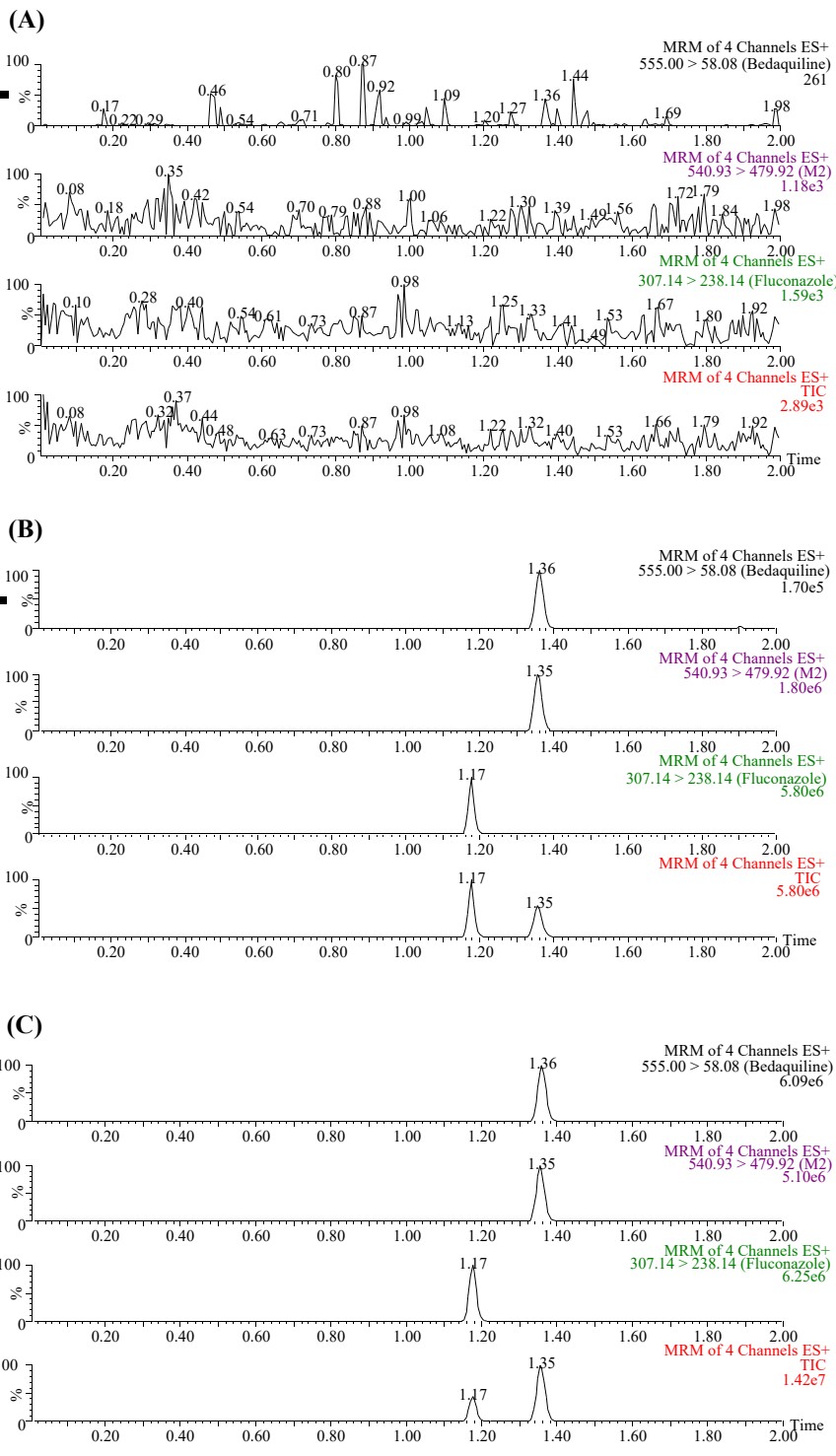

**Figure 1  Typical UPLC-MS/MS chromatograms of bedaquiline, M2, and IS (fluconazole).** (A) Blank plasma sample. (B) Blank plasma samples were spiked with 10 ng/mL bedaquiline, 10 ng/mL M2, and 200 ng/mL IS. (C) Sprague-Dawley rat plasma samples 4 h after oral administration of bedaquiline.

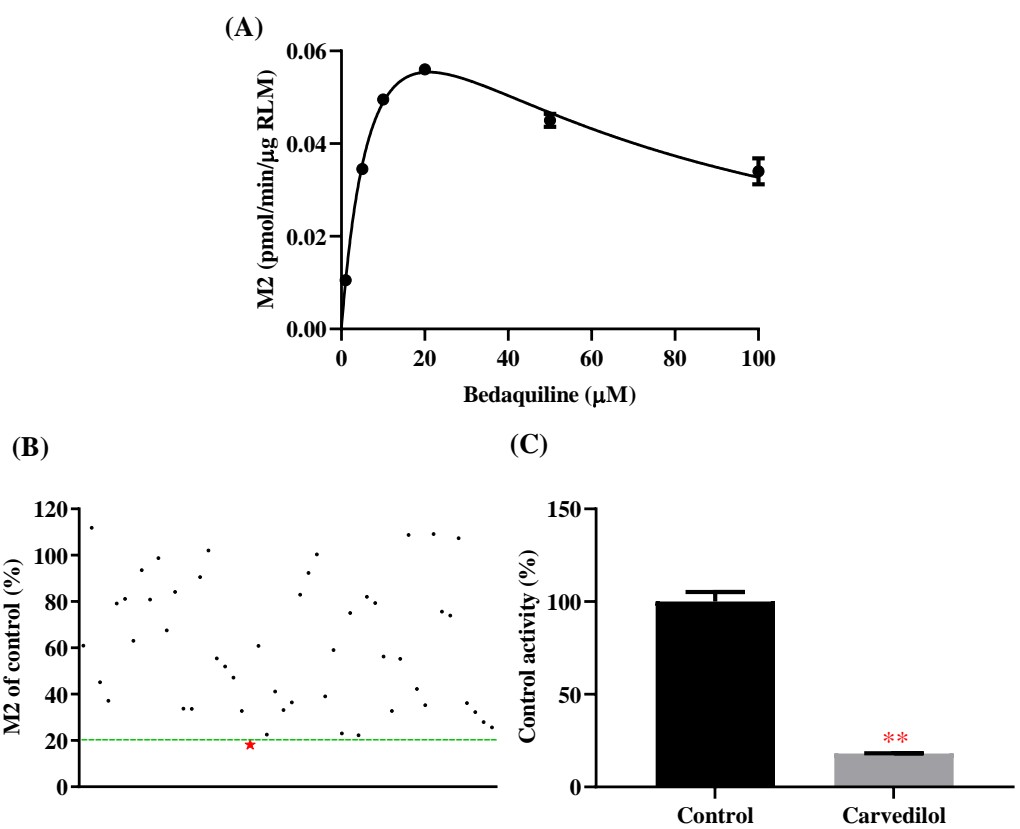

**Figure 2** **Potential DDIs of 50 varieties of drugs with bedaquiline in RLM.** (A) Michaelis–Menten curve of bedaquiline. (B) Inhibitory effects of 50 drugs (100 μM) on bedaquiline metabolism. (C) Inhibition of bedaquiline by carvedilol. Data are expressed as mean ± SD.

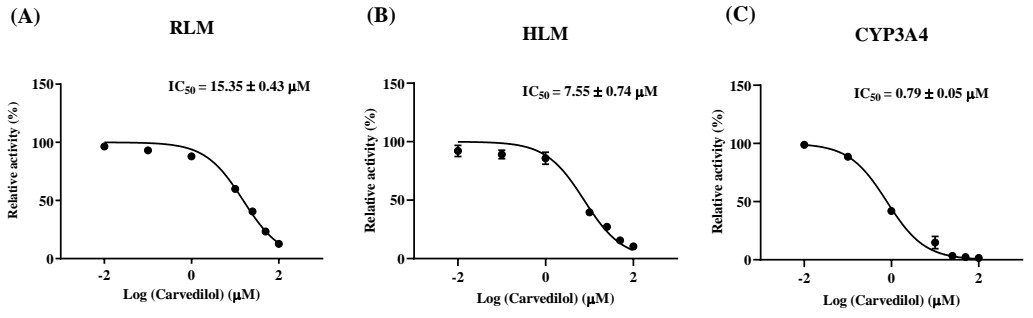

**Figure 3** **The inhibitory strength of carvedilol to bedaquiline.** IC$_{50}$ curves of carvedilol with diverse concentrations in RLM (A), HLM (B), and CYP3A4 (C). Data are expressed as the mean ± SD, $n = 3$.

year, is ranked as the second leading cause of death worldwide among infectious diseases (*Chan, Khadem & Brown, 2013*; *Nguyen et al., 2016*; *Fox & Menzies, 2013*). MDR-TB is resistant to first-line anti-TB drugs such as rifampicin and isoniazid (*Deshkar & Shirure,*

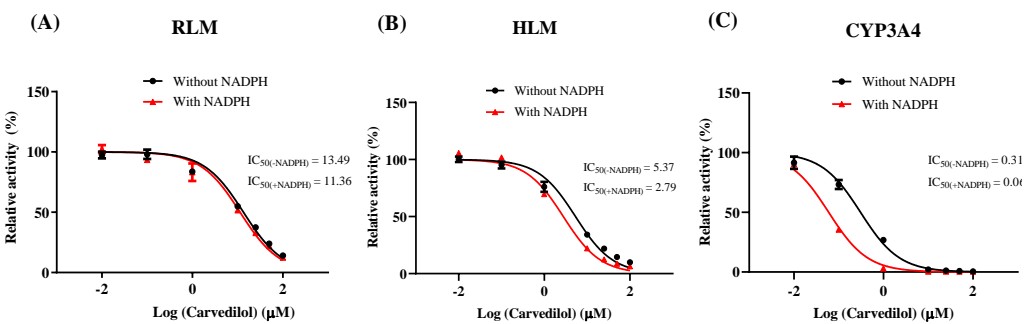

**Figure 4 Determination the time-dependent inhibitor of carvedilol to bedaquiline.** $IC_{50}$ shift curves of carvedilol on RLM (A), HLM (B), and CYP3A4 (C) with or without NADPH. Data are expressed as the mean $\pm$ SD, $n = 3$.

*2022*) and has received growing global concern as rising levels of drug resistance lead to skyrocketing death rates (*Kotwal et al., 2020*).

Bedaquiline, a diarylquinoline compound, is the only drug approved by the FDA for the treatment of MDR-TB in 2012 (*Mahajan, 2013*; *Preiss et al., 2015*) and has a significant inhibitory effect on various non-tuberculous mycobacteria, such as Mycobacterium avium, Mycobacterium ulcerans, and Mycobacterium abscessus (*Andries et al., 2005*). Bedaquiline is a *p*-gp substrate and undergoes phase I metabolism in humans, mainly through CYP3A4 in the liver to N-monomethylated metabolite M2 (*Kotwal et al., 2020*; *Lakshmanan & Xavier, 2013*). In this study, we screened a series of drugs to investigate potential DDIs between bedaquiline and drugs for cardiovascular diseases, such as carvedilol, medications for gastrointestinal disorders, such as cimetidine, and traditional Chinese medicine with anti-inflammatory properties, such as resveratrol.

According to the results, carvedilol has the most potent restraining effect on bedaquiline. Carvedilol is used to treat heart failure and hypertension by blocking the $\beta1$, $\beta2$, and $\alpha$-1 adrenergic receptors (*Gilbert et al., 1996*). Compared with other heart failure drugs, such as metoprolol, it has more beneficial effects, such as remodeling and central hemodynamics (*Gilbert et al., 1996*; *Di Lenarda et al., 1999*; *Metra et al., 2000*; *Sanderson et al., 1999*). Additionally, carvedilol is a substrate of *p*-gp (*Brodde & Kroemer, 2003*), and although it is not metabolized primarily by CYP3A4, it has been demonstrated to participate in carvedilol's metabolism (*Oldham & Clarke, 1997*; *Iwaki et al., 2018*; *Iwaki et al., 2016*). Consequently, a vital conjecture was that there might be a DDI between bedaquiline and carvedilol.

*In vitro* experiments, the $IC_{50}$ of carvedilol to bedaquiline was $15.35 \pm 0.43$ μM in RLM, $7.55 \pm 0.74$ μM in HLM, and $0.79 \pm 0.05$ μM in CYP3A4, with the inhibition type of mixed, un-competitive, and mixed, respectively. *In vivo* experiments, compared with the single group, the $AUC_{(0-t)}$, $AUC_{(0-\infty)}$, and $C_{max}$ values of bedaquiline went up by 2.15–2.28 fold in the combined group and accompanied by a 4.33-fold reduction in $CL_{z/F}$. These pharmacokinetic parameters alterations indicated that carvedilol significantly enhanced the plasma exposure and prolonged the accumulation duration of bedaquiline

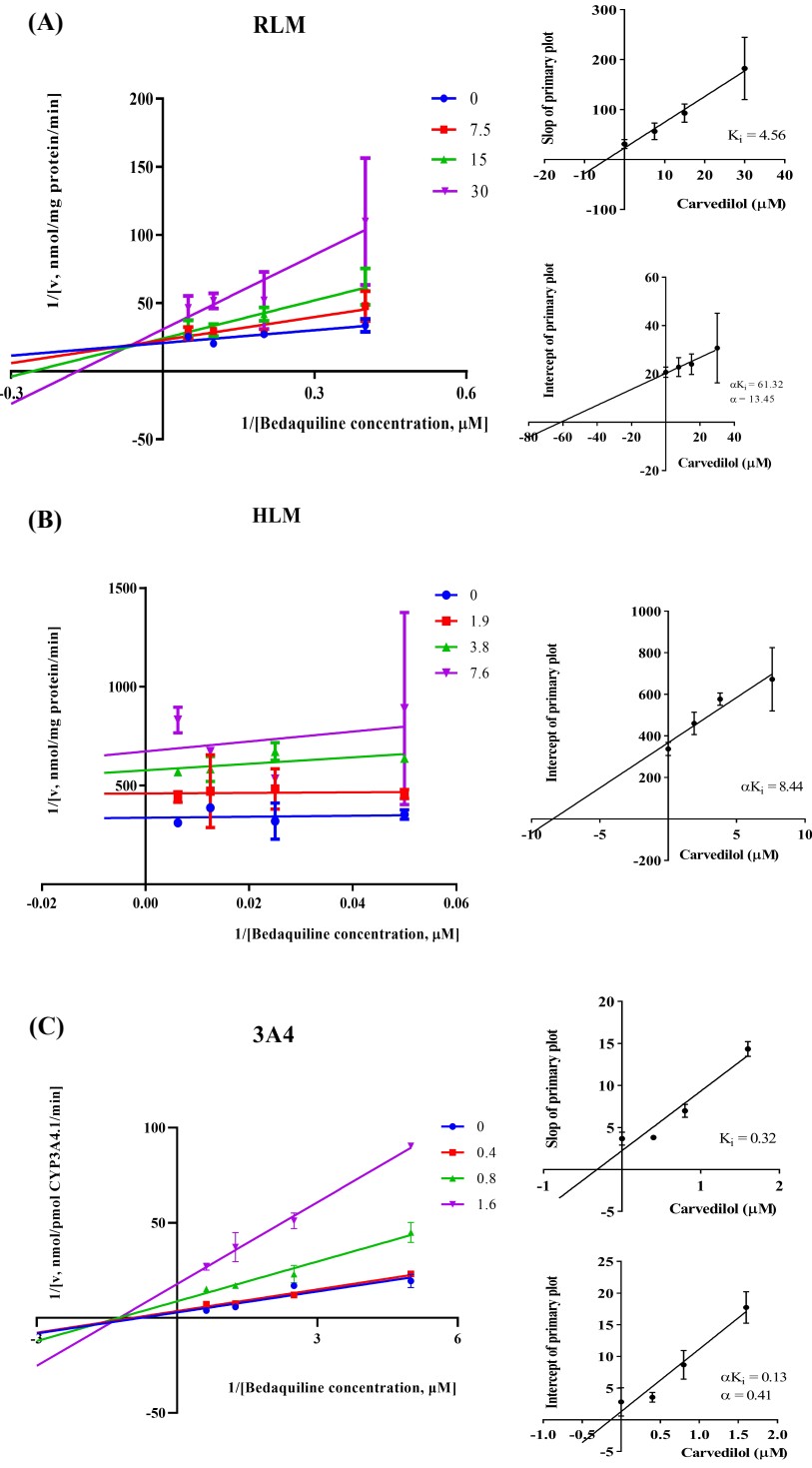

**Figure 5** **The inhibition type of carvedilol to bedaquiline.** A mixed type of non-competitive + competitive inhibition in RLM (A), an un-competitive inhibition in HLM (B), and a mixed type of non-competitive + un-competitive inhibition in CYP3A4 (C). Data are expressed as the mean ± SD.

**Table 1 Effect of carvedilol on inhibiting bedaquiline metabolism in RLM/HLM/CYP3A4 and corresponding $IC_{50}$ values.**

|  | Inhibition type | Ki ($\mu$M) | $\alpha$Ki ($\mu$M) | $\alpha$ | $IC_{50}$ values ($\mu$M) |
|---|---|---|---|---|---|
| RLM | Mixed | 4.56 | 61.32 | 13.45 | $15.35 \pm 0.43$ |
| HLM | Un–Competitive | – | 8.44 | – | $7.55 \pm 0.74$ |
| CYP3A4 | Mixed | 0.32 | 0.13 | 0.41 | $0.79 \pm 0.05$ |

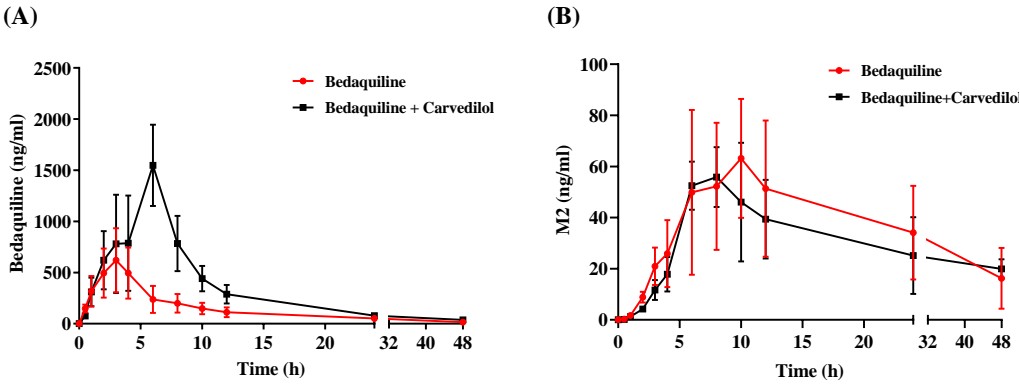

**Figure 6 Mean concentration–time curves of bedaquiline (A) and M2 (B).** Mean concentration–time curves of bedaquiline (A) and M2 (B) in the single group (bedaquiline alone) and the combined group (bedaquiline and carvedilol). Data are expressed as the mean $\pm$ SD, $n = 4$.

in rats, potentially elevating the risk of bedaquiline-associated adverse reactions, including cardiotoxicity (*Worley & Estrada, 2014*), hepatotoxicity (*Kakkar & Dahiya, 2014*), and phospholipidosis (*Diacon et al., 2012*; *Guillemont et al., 2011*). The saturation of *p-gp* transport likely explains the dose-dependent increase of bedaquiline (*Martín-García & Esteban, 2021*). However, the main pharmacokinetic parameters of M2 did not have any obvious variation, and the AUC increase in bedaquiline may not be mediated by carvedilol's effects on CYP3A4 or *p-gp*, which needed further research. Moreover, when carvedilol was co-administered with bedaquiline, it prolonged the $T_{max}$ of bedaquiline while shortening its $t_{1/2z}$. Elimination half-time refers to the time taken for the drug concentration in the body to decrease to half of its initial dose. Typically, the duration of drug action, time to reach a steady-state level, and clearance time are all influenced by the drug's $t_{1/2z}$. Generally, the time for the drug to reach steady-state concentrations *in vivo* is 4–5 $t_{1/2z}$ (*Andrade, 2022*), suggesting that carvedilol may reduce the time for bedaquiline to reach steady-state plasma concentrations.

However, DDIs and gene polymorphisms of metabolic enzymes are the main factors causing otherness in clinical drug plasma exposure. In this study, we only analyzed DDIs; however, CYP3A4 enzyme activity varies widely between individuals (up to 60-fold), leading to treatment failure or unforeseeable toxicity and side effects (*Hu et al., 2017*). In future studies, we expect to study the gene polymorphisms of its metabolic enzymes to

**Table 2  Main pharmacokinetic parameters of bedaquiline in two groups of rats ($n = 4$).**

| Parameters | Bedaquiline (10 mg/kg) | Bedaquiline (10 mg/kg) + Carvedilol (5 mg/kg) |
|---|---|---|
| $AUC_{(0-t)}$ (ng/mL*h) | $5{,}250.60 \pm 1{,}693.23$ | $11{,}651.25 \pm 3{,}609.65$[*] |
| $AUC_{(0-\infty)}$ (ng/mL*h) | $5{,}510.31 \pm 1{,}794.78$ | $11{,}835.57 \pm 3{,}671.52$[*] |
| $t_{1/2z}$ (h) | $12.00 \pm 1.01$ | $7.15 \pm 2.71$[*] |
| $T_{max}$ (h) | $2.00 \pm 0.82$ | $5.25 \pm 1.50$[**] |
| $V_{z/F}$ (L/kg) | $68.18 \pm 23.18$ | $9.07 \pm 3.05$[***] |
| $CL_{z/F}$ (L/h/kg) | $3.94 \pm 1.28$ | $0.91 \pm 0.26$[***] |
| $C_{max}$ (ng/mL) | $639.61 \pm 273.93$ | $1{,}458.70 \pm 370.54$[*] |

Notes.

AUC, area under the plasma concentration–time curve; $t_{1/2z}$, elimination half time; $T_{max}$, peak time; $V_{z/F}$, apparent volume of distribution; $CL_{z/F}$, plasma clearance; $C_{max}$, maximum plasma concentration.

[*] $P < 0.05$.

[**] $P < 0.01$.

[***] $P < 0.005$ in comparison with the single group.

**Table 3  Main pharmacokinetic parameters of M2 in two groups of rats ($n = 4$).**

| Parameters | Bedaquiline (10 mg/kg) | Bedaquiline (10 mg/kg) + Carvedilol (5 mg/kg) |
|---|---|---|
| $AUC_{(0-t)}$ (ng/mL*h) | $1{,}550.69 \pm 831.88$ | $1{,}238.59 \pm 641.31$ |
| $AUC_{(0-\infty)}$ (ng/mL*h) | $2{,}133.95 \pm 1{,}376.90$ | $1{,}886.52 \pm 938.03$ |
| $t_{1/2z}$ (h) | $19.68 \pm 1.22$ | $18.44 \pm 4.57$ |
| $T_{max}$ (h) | $33.69 \pm 9.11$ | $39.63 \pm 8.77$ |
| $V_{z/F}$ (L/kg) | $21.22 \pm 6.53$ | $25.32 \pm 4.80$ |
| $CL_{z/F}$ (L/h/kg) | $9.00 \pm 2.58$ | $10.00 \pm 1.63$ |
| $C_{max}$ (ng/mL) | $176.37 \pm 72.07$ | $258.09 \pm 178.02$ |

Notes.

AUC, area under the plasma concentration–time curve; $t_{1/2z}$, elimination half time; $T_{max}$, peak time; $V_{z/F}$, apparent volume of distribution; $CL_{z/F}$, plasma clearance; $C_{max}$, maximum plasma concentration.

provide medical workers with more comprehensive experimental data on the correct use of bedaquiline.

# CONCLUSIONS

Carvedilol inhibited the metabolism of bedaquiline *in vitro* with mixed, un-competitive, and mixed mechanisms in RLM, HLM, and CYP3A4, respectively, and significantly changed the main pharmacokinetic parameters of bedaquiline *in vivo* to increase its plasma exposure. As a consequence, the combination of bedaquiline and carvedilol should be avoided, or regular plasma concentration monitoring should be performed to reduce the severity and frequency of bedaquiline-related side effects.

## Abbreviations

| | |
|---|---|
| **ABC** | ATP-binding cassette |
| **ACN** | acetonitrile |
| **AUC** | area under the plasma concentration-time curve |
| **$CL_{z/F}$** | plasma clearance |

| $C_{max}$ | maximum plasma concentration |
|---|---|
| **CMC-Na** | carboxymethylcellulose sodium salt |
| **CVS** | cardiovascular system |
| **CYP450** | cytochrome P450 |
| **CYP3A4** | cytochrome P450 3A4 |
| **DAS** | Drug and statistics |
| **DDIs** | drug–drug interactions |
| **FDA** | Food and Drug Administration |
| **HF** | heart failure |
| **HLM** | human liver microsomes |
| **IACUC** | Institutional Animal Care and Use Committee |
| $IC_{50}$ | the half-maximal inhibitory concentration |
| **IS** | internal standard |
| $K_i$ | inhibition constant |
| $K_m$ | Michaelis–Menten constant |
| **LLOQ** | the lower limit of quantitation |
| **ME** | matrix effect |
| **MDR-TB** | multidrug-resistant tuberculosis |
| **NADPH** | Reduced nicotinamide adenine dinucleotide phosphate |
| **NTM** | non-tuberculous mycobacteria |
| **PBS** | phosphate buffered saline |
| *P*-**gp** | *P*-glycoprotein |
| **RLM** | rat liver microsomes |
| **SD** | standard deviation |
| **SD rats** | Sprague-Dawley rats |
| **SRM** | selective response monitoring |
| **TCM** | Traditional Chinese Medicine |
| **TDI** | time-dependent inhibition |
| **TIC** | total ionic chromatography |
| $t_{1/2z}$ | elimination half time |
| $T_{max}$ | peak time |
| **UPLC-MS/MS** | ultra-performance liquid chromatography tandem mass spectrometry |
| $V_{z/F}$ | apparent volume of distribution |

## ACKNOWLEDGEMENTS

We appreciated Xu Ren-ai, chief pharmacist of the First Affiliated Hospital of Wenzhou Medical University for providing us with their experimental platform and instruments to support this work.

### Funding

This work was supported by Ningbo Medical & Health Leading Academic Discipline Project (No. 2022-Z03), Ningbo Medical & Health Brand Discipline Project (No. PPXK2024-08). The APC was funded by the Ningbo Traditional Chinese Medicine Pharmaceutical Center Construction Project (Project No. zyy23011) and the corresponding author Hangjuan Lin is the project funder. The funders had no role in study design, data collection and analysis, decision to publish, or preparation of the manuscript.

### Grant Disclosures

The following grant information was disclosed by the authors:
Ningbo Medical & Health Leading Academic Discipline Project: No. 2022-Z03.
Ningbo Medical & Health Brand Discipline Project: No. PPXK2024-08.
APC: Traditional Chinese Medicine Pharmaceutical Center Construction Project: No. zyy23011.

### Competing Interests

The authors declare no known competing interests or personal relationships that could have appeared to influence the work reported in this paper.

### Author Contributions

- Qingqing Li conceived and designed the experiments, performed the experiments, analyzed the data, prepared figures and/or tables, and approved the final draft.
- Wanshu Li conceived and designed the experiments, performed the experiments, analyzed the data, prepared figures and/or tables, and approved the final draft.
- Jie Chen performed the experiments, analyzed the data, prepared figures and/or tables, and approved the final draft.
- Hangjuan Lin conceived and designed the experiments, authored or reviewed drafts of the article, and approved the final draft.
- Cixia Zhou conceived and designed the experiments, authored or reviewed drafts of the article, and approved the final draft.

### Animal Ethics

The following information was supplied relating to ethical approvals (i.e., approving body and any reference numbers):

The study protocol followed the ARRIVE guidelines and was approved by the Laboratory Animal Ethics Committee of The First Affiliated Hospital of Wenzhou Medical University (Ethics approval number: WYYY-IACUC-AEC-2023-046).

### Data Availability

The raw measurements are available in the Supplementary File.

## Supplemental Information

Supplemental information for this article can be found online at http://dx.doi.org/10.7717/peerj.19313#supplemental-information.

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
