# Peer review of "Inhibitory effect of carvedilol on bedaquiline metabolism in vitro and in vivo"

_PeerJ, doi:10.7717/peerj.19313_

## Round 0.1 · original submission · Major Revisions

Dear authors,

Manuscript titled "Inhibitory effect of carvedilol on bedaquiline metabolism in vitro and in vivo" that you submitted to PeerJ has been reviewed.
The reviewer(s) have suggested that some important points must be clarified and have requested substantial changes to be made in the manuscript. Therefore, I invite you to respond to the reviewer(s)' comments and revise your manuscript. The reviewer(s) comments are included at the end of this letter.

Please ensure that all review, editorial, and staff comments are addressed in a response letter and that any edits or clarifications mentioned in the letter are also inserted into the revised manuscript where appropriate.

Reviewer 1 ·

Basic reporting

This manuscript investigates the effect of carvedilol on bedaquiline metabolism in liver microsomes and rats. The study findings indicate that carvedilol inhibits M2 formation in rat and human liver microsomes as well as in CYP3A4 andcarvedilol increases the area under the curve (AUC) of bedaquiline in rats. The following concerns and comments are listed as below:
1. Role of P-gp and CYP3A4: Carvedilol is both a substrate and a weak inhibitor of P-glycoprotein (P-gp). As discussed, the saturation of P-gp transporters likely explains the dose-dependent increase in bedaquiline levels. Carvedilol inhibits CYP3A4, as shown in in vitro studies. Therefore, the pharmacokinetic (PK) study should include the PK profile of the M2 metabolite. This data would help clarify whether the AUC increase in bedaquiline is mediated by carvedilol’s effects on CYP3A4 or P-gp.
2. Half-life Changes: The co-administration of carvedilol and bedaquiline results in a shorter half-life compared to bedaquiline alone. This observation requires further discussion in the manuscript.
3. The co-administration of carvedilol and bedaquiline is not common in clinical practice.
4. In Figure 2B, three additional compounds are shown to inhibit M2 formation by approximately 80%. The rationale for excluding these compounds from further investigation should be clarified.
5. The justification for the doses of carvedilol and bedaquiline used in the PK study is required. Additionally, the rationale for selecting specific concentrations of bedaquiline (10 µM in rat liver microsomes, 80 µM in human liver microsomes, and 0.8 µM in CYP3A4) for time-dependent inhibition (TDI) studies should be explained.
6. English should be polished.

Experimental design

See section 1

Validity of the findings

See section 1

Additional comments

See section 1

Reviewer 2 ·

Basic reporting

This manuscript studied the inhibitory effect of carvedilol on bedaquiline metabolism, and the results showed that Carvedilol could obviously inhibit the metabolism of bedaquiline both in vitro and in vivo. The manuscript is well-designed, the method is feasible, and the conclusion is correct.
There are two minor issues with the manuscript:
1. The drugs listed from lines 107 to 114, which were not involved in the experiment, can be deleted.
2. From Figure 1, it can be seen that M2 was detected in the plasma. Why are there no pharmacokinetic parameters and drug time curves for M2?

Experimental design

The experimental design is reasonable.

Validity of the findings

The findings confirm drug drug interactions and provide a theoretical basis for clinical medication.

Reviewer 3 ·

Basic reporting

The article is clear but some of the grammar and word choice should be revised for further clarity for readers.

Experimental design

The results presented are logical and follow a verifiable hypothesis.

Validity of the findings

The data, discussion, and conclusions are strong and provide additional insights into the use of bedaquiline.

Additional comments

A detailed review per section of the article is attached to this report.

Annotated reviews are not available for download in order to protect the identity of reviewers who chose to remain anonymous.

---

## Round 0.2 · accepted · Accept

Dear Author,

Congratulations, after the good work of revisions in response to the reviewers' comments, I would like to inform you that your manuscript has been accepted for publication in PeerJ.

Reviewer 2 ·

Basic reporting

no comment

Experimental design

no comment

Validity of the findings

no comment

Reviewer 3 ·

Basic reporting

The article is clear and shows interesting results.

Experimental design

The results presented are logical and follow a verifiable hypothesis.

Validity of the findings

The data, discussion, and conclusions are strong and provide additional insights into the use of bedaquiline.

Additional comments

The authors addressed my previous concerns satisfactorily, and I don't see any further issues. I think the article is ready for publication now.